



# Predicting and correcting the influence of boundary conditions in regional inverse analyses

Hannah Nesser[1,2], Kevin W. Bowman[1], Matthew D. Thill[1], Daniel J. Varon[2], Cynthia Randles[3*], Ashutosh Tewari[3+], Felipe J. Cardoso-Saldaña[3], Emily Reidy[3], Joannes D. Maasakkers[4], Daniel J. Jacob[2]

[1]Jet Propulsion Laboratory, California Institute of Technology, Pasadena, CA, USA
[2]Harvard University School of Engineering and Applied Sciences, Cambridge, MA, USA
[3]ExxonMobil Technology and Engineering Company, Annandale, NJ, USA
[4]SRON Space Research Organization Netherlands, Leiden, the Netherlands
[*]Now at Scepter, Inc., San Francisco, CA, USA
[+]Now at Amazon Supply Chain Optimization Technologies, Seattle, WA, USA

*Correspondence to*: Hannah Nesser (hannah.o.nesser@jpl.nasa.gov)

**Abstract.** Regional inverse analyses of atmospheric trace gas observations quantify gridded two-dimensional surface fluxes by fitting the observations to simulated concentrations from a chemical transport model (CTM), usually by Bayesian optimization regularized by a gridded prior flux estimates. Regional inversions rely on the specification of background concentrations given by the boundary conditions (BCs) at the edges of the inversion domain, but biases in the BCs propagate to biases in the optimized fluxes. We develop a theoretical framework to explain how errors in the BCs influence the optimized fluxes as a function of the prior and observing system error statistics and of CTM transport. We derive a preview metric to estimate the BC-induced errors before conducting an inversion to support domain specification and a diagnostic metric to accurately quantify these errors after solving the inversion. We compare two methods to correct BC biases as part of an inversion, either directly by optimizing BC concentrations (boundary method) or indirectly by correcting grid cell fluxes outside the domain of interest (buffer method). We demonstrate that the boundary method is generally more accurate, physically grounded, and computationally tractable.

## 1 Introduction

Regional inversions of observed atmospheric trace gas concentrations quantify surface fluxes by fitting simulated concentrations from a chemical transport model (CTM) to the observations assuming background concentrations given by boundary conditions (BCs) at the edge of the inversion domain. Such analyses can improve knowledge of fluxes and their trends on local to continental scales at high spatiotemporal resolution while avoiding the need to accurately quantify fluxes and concentrations globally (Sargent et al., 2021; Nesser et al., 2024; Byrne et al., 2024). However, BC concentrations are





often uncertain and biases in the BCs propagate to the inferred fluxes. Here we examine the problem of how BC biases affect regional inversions of greenhouse gas fluxes and develop a framework to predict and correct the influence of these biases.

The BCs used to define background concentrations can be provided by coarse-resolution global simulations (Göckede et al., 2010; Wecht et al., 2014), by trace gas observations at the domain edge (Lauvaux et al., 2016; Balashov et al., 2020), or by combining simulated and observed concentrations (Sargent et al., 2018; Estrada et al., 2024). The information used to constrain BCs is often spatiotemporally sparse and may be biased. Even small BC biases can result in large biases in the inferred fluxes (Göckede et al., 2010; Lauvaux et al., 2012; Karion et al., 2021). BC biases are particularly critical for inverse analyses of long-lived gases such as carbon dioxide and methane where concentrations and variability within the domain of interest can depend significantly on inflow concentrations.

Regional inversions generally infer gridded surface fluxes (the state vector) by minimizing a Bayesian cost function that accounts for the error statistics of the observing system (including the observations and CTM) and of the prior flux estimate used to regularize the solution. BC biases may be corrected as part of the inversion by optimizing BC concentrations as part of the state vector (Lauvaux et al., 2012; Wecht et al., 2014), by allowing buffer grid cells at the edge of the domain to absorb BC biases with potentially unphysical flux corrections (Shen et al., 2021; Varon et al., 2022), or by combining these approaches (Estrada et al., 2024). Buffer grid cells increase the inversion's domain and, therefore, its computational cost.

We present here a theoretical framework to predict, diagnose, and correct the influence of BC biases on the posterior gridded surface fluxes generated by a regional inversion (Sect. 2). We consider an inert trace gas with no sources or sinks within the inversion domain other than the surface fluxes. We demonstrate this framework with increasingly complex one- and two-dimensional simulation experiments that invert pseudo-observations of a long-lived trace gas generated using known fluxes and BCs (Sects. 3 and 4, respectively). Throughout, we compare different methods to correct BC biases within inversions and demonstrate metrics that estimate the effect of BC biases.

## 2 Quantifying the effect of boundary condition errors

We describe the analytical inverse solution, which we use to derive an exact solution (diagnostic) for the sensitivity of the posterior fluxes to a BC bias (Sect. 2.1). We apply this diagnostic to a simple one-dimensional transport model to determine how these errors depend on the inversion parameters (Sect. 2.2). We then derive a simplified version of the diagnostic (preview) to predict the sensitivity of the posterior fluxes to BC biases before conducting the inversion to support domain specification (Sect. 2.3).



## 2.1 Diagnostic equation

Given an $n$-dimensional state vector of gridded fluxes $\boldsymbol{x}$ and an $m$-dimensional vector of observations $\boldsymbol{y}$, both with normally distributed errors, the optimal flux estimate $\hat{\boldsymbol{x}}$ is obtained by minimizing a Bayesian cost function

$$J(\boldsymbol{x}) = (\boldsymbol{x} - \boldsymbol{x}_A)^T \mathbf{S}_A^{-1}(\boldsymbol{x} - \boldsymbol{x}_A) + \left(\boldsymbol{y} - F(\boldsymbol{x})\right)^T \mathbf{S}_O^{-1}\left(\boldsymbol{y} - F(\boldsymbol{x})\right), \tag{1}$$

where $\boldsymbol{x}_A$ and $\mathbf{S}_A$ are the prior flux estimate and error covariance matrix, respectively, $\mathbf{S}_O$ is the observing system error covariance matrix representing uncertainties in the observations and model, and $F$ is the CTM (Brasseur and Jacob, 2017). We assume as is standard that the CTM initial conditions are given by a sufficiently long spin-up simulation driven by the BCs so that the initial conditions are unbiased compared to the BCs. If the CTM is linear, then $F(\boldsymbol{x}) = \mathbf{K}\boldsymbol{x} + \boldsymbol{c}$ where $\mathbf{K} = \partial F / \partial \boldsymbol{x}$ is the Jacobian matrix. The vector $\boldsymbol{c}$ represents the model background defined by the transport of the BCs to the same spatiotemporal locations as the observations. If the BC concentrations are optimized as part of the inversion, information about the background is instead contained in the state vector and in the columns of the Jacobian matrix so that $\boldsymbol{c} = \mathbf{0}$.

We can write the analytical solution for the cost function minimum that yields the optimal (posterior) flux estimate $\hat{\boldsymbol{x}}$ as

$$\hat{\boldsymbol{x}} = \boldsymbol{x}_A + \mathbf{G}\big(\mathbf{y} - (\mathbf{K}\mathbf{x}_A + \boldsymbol{c})\big) \tag{2}$$

where

$$\mathbf{G} = \frac{\partial \hat{\boldsymbol{x}}}{\partial \boldsymbol{y}} = (\mathbf{K}^T \mathbf{S}_O^{-1}\mathbf{K} + \mathbf{S}_A^{-1})^{-1}\mathbf{K}^T \mathbf{S}_O^{-1} \tag{3}$$

is the gain matrix that represents the sensitivity of the posterior fluxes to the observations. We separate the model-observation difference into the contributions from the errors in the prior fluxes relative to the true fluxes ($\boldsymbol{x}_T$) and the errors in the observing system ($\boldsymbol{\varepsilon}$) so that

$$\hat{\boldsymbol{x}} = \boldsymbol{x}_A + \mathbf{A}(\boldsymbol{x}_T - \boldsymbol{x}_A) + \mathbf{G}\boldsymbol{\varepsilon} \tag{4}$$

where $\mathbf{A} = \partial \hat{\boldsymbol{x}} / \partial \boldsymbol{x} = \mathbf{G}\mathbf{K}$ is the averaging kernel matrix, a measure of inversion information content that gives the sensitivity of the posterior fluxes to the true fluxes. The trace of the averaging kernel matrix gives the degrees of freedom for signal (DOFS), the number of independent pieces of information quantified by the inversion (Rodgers, 2000).



The posterior error induced by a BC error ($\boldsymbol{\varepsilon}_\mathrm{C}$) is derived by comparing the posterior fluxes produced with the true BC ($\boldsymbol{c}_\mathrm{T}$) to an inversion with the BC error ($\boldsymbol{c} = \boldsymbol{c}_\mathrm{T} + \boldsymbol{\varepsilon}_\mathrm{C}$):

$$\Delta\widehat{\boldsymbol{x}} = -\mathbf{G}\boldsymbol{\varepsilon}_\mathrm{C}. \tag{5}$$

We define Eq. (5) as the BC diagnostic that estimates the effect of BC bias on the posterior fluxes given assumed BC error statistics and an explicitly constructed Jacobian matrix. BC-induced posterior errors are controlled by the gain matrix, which is a function of the observing system errors, prior flux errors, and transport as represented by the Jacobian matrix.

Equation (5) does not account for inverse methods used to correct for BC biases. We compare the effect of optimizing the BC concentrations (boundary method) or fluxes outside of the domain of interest (buffer method). The boundary method optimizes one or more terms corresponding to BC concentrations along with the domain fluxes (Lauvaux et al., 2012; Wecht et al., 2014; Hancock et al., 2025). The buffer approach expands the inversion domain to optimize fluxes in outlying grid cells, which are allowed to vary unphysically to absorb BC biases and are therefore excluded from the final analysis (Shen et al., 2021; Varon et al., 2022). Both approaches increase the state vector dimension, thereby altering the Jacobian matrix and prior error covariance matrix. The buffer approach also applies large prior error standard deviations to the buffer grid cells, often by aggregating the buffer grid cells into large clusters to minimize the increase in the state vector dimension.

We compare the posterior fluxes generated by an inversion with the true BC and no correction method to the posterior fluxes generated by an inversion with a perturbed BC and an error correction method. The increase in the state vector dimension resulting from the error correction method results in the addition of rows to the gain matrix (Eq. (3)). We define the reduced gain matrix including only the rows corresponding to the state vector elements within the domain of interest as $\mathbf{G}'$, which has the same dimension as the gain matrix for the inversion with no correction method ($\mathbf{G}$). The matrix and vector elements associated with the non-domain state vector elements (BC concentrations or buffer grid cells) are denoted with a positive superscript ($^+$). The difference between the posterior fluxes is then:

$$\Delta\widehat{\boldsymbol{x}} = (\mathbf{A}' - \mathbf{A})(\boldsymbol{x}_\mathrm{T} - \boldsymbol{x}_\mathrm{A}) + \mathbf{A}'^{+}(\boldsymbol{x}_\mathrm{T}^{+} - \boldsymbol{x}_\mathrm{A}^{+}) + (\mathbf{G}' - \mathbf{G})\boldsymbol{\varepsilon}_\mathrm{C} \tag{6}$$

where $\mathbf{A}' = \mathbf{G}'\mathbf{K}$ and $\mathbf{A}'^{+} = \mathbf{G}'\mathbf{K}^{+}$ are the averaging kernel matrices for the corrected inversion associated with the domain and non-domain state vector elements, respectively. The first two terms of Eq. (6) represent the change in information content resulting from the correction method while the third term represents the change in the influence of the BC bias. The true – prior flux difference ($\boldsymbol{x}_\mathrm{T} - \boldsymbol{x}_\mathrm{A}$) is unknown but can be approximated by the prior flux standard deviations.



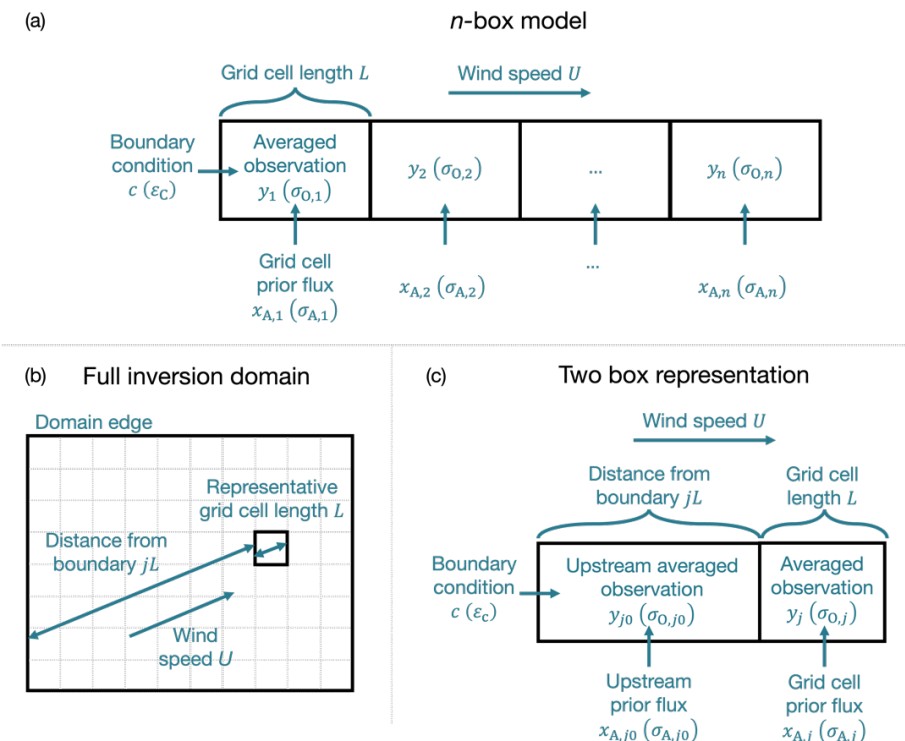

**Figure 1: One-dimensional and two-box models of a passive trace gas to quantify the influence of BCs on inverse analyses. The one-dimensional model (a) simulates the concentrations of an inert trace gas over $n$ grid cells of length $L$ given a prescribed BC $c$, fluxes $x = [x_1, x_2, \ldots, x_n]^{\mathrm{T}}$, and advection with wind speed $U$. The two-box model (c) generalizes an inverse analysis optimizing fluxes over a two-dimensional grid (b) where all grid cells have a representative grid cell length $L$ given the wind direction. Each grid cell in the domain is reduced to a two-box model composed of a cluster of the $j$ upstream grid cells (index $j0$) and the grid cell itself (index $j$). In all models, the inversion is solved with the averaged observation $y_i$ over each grid cell $i = \{1, \ldots, n\}$ with corresponding error standard deviation $\sigma_{O,i}$. The inversion is regularized by the prior flux $x_{A,i}$ in each grid cell with corresponding prior error standard deviation $\sigma_{A,i}$.**

## 2.2 One-dimensional model

To better understand how observing system errors, prior flux errors, and transport determine the influence of BC biases on posterior flux estimates (Eq. (5)), we consider a one-dimensional horizontal transport model with constant wind speed $U$. Figure 1 (panel a) depicts this model for $n$ grid cells of length $L$. The Jacobian matrix for this transport model is derived through steady-state mass balance (Supplement S1) to be lower diagonal with constant values given by

$$\tau = \alpha \tau' \tag{7}$$

where $\alpha$ converts between the flux units and the observation units and $\tau'$ is the grid cell residence time calculated as $\tau' = L/U$. The trace gas sources in the model are the BC $c$ and fluxes $x = [x_1, x_2, \ldots, x_n]^{\mathrm{T}}$. We assume that the $m$ observations



are uniformly distributed in space and time with constant uncorrelated error variance $\sigma_O^2$. We solve the inversion (Eq. (2)) using the averaged observations in each grid cell so that the observing system error covariance matrix $\mathbf{S}_O$ is diagonal with constant error variances $\sigma_O^2/m_g$ where $m_g = m/n$ is the number of observations in each grid cell. We assume constant prior fluxes $x_A$ and diagonal prior error covariance matrix $\mathbf{S}_A$ with constant error variances $\sigma_A^2$.

We define the dimensionless, domain-average information ratio $R$ of the prior error variances in concentration units to the observing system error variances

$$R = m_g \left(\frac{\tau \sigma_A}{\sigma_O}\right)^2. \tag{8}$$

The information ratio increases with decreasing observing system error standard deviation, increasing prior error standard deviation, and increasing residence time so that the observations are more sensitive to the fluxes relative to the BC. Large values of the information ratio ($R \gg 1$) represent the case where the prior errors are larger than the observing system errors so that the posterior fluxes are strongly constrained by the observations (observation-rich). Small values ($R \ll 1$) correspond to inversions that are limited by the number or uncertainty of the observations (observation-limited), including the common 160 case of spatially heterogeneous observations. Consider an illustrative inversion of a methane-like trace gas with $\tau = 1.4$ h (corresponding to $U = 5$ m s$^{-1}$ and $L = 25$ km), $\sigma_A = 12.5$ ppb h$^{-1}$ (corresponding to 50% uncertainty on relatively large prior emissions of 25 ppb h$^{-1}$), and $\sigma_O = 10$ ppb. In this case, an average of $m_g = 200$ observations per grid cell are needed to achieve $R = 1$. The information ratio for an inversion can be increased by increasing the inversion duration to include more observations or by coarsening the resolution of the gridded fluxes optimized by the inversion.

The effect of a constant BC bias $\varepsilon_C$ on the posterior fluxes is calculated as a function of the information ratio with the diagnostic (Eq. (5)). We derive (Supplement S2) the BC-induced error in the limiting cases of inversions that are observation-limited ($R \ll 1$) or observation-rich ($R \gg 1$) :

$$\Delta \hat{x}_j = \begin{cases} -(\tau^{-1}\varepsilon_C)(n-j+1)R, & R \ll 1 \\ -(\tau^{-1}\varepsilon_C)R^{-j+1}, & R \gg 1 \end{cases}, \quad j = \{1, \ldots, n\}. \tag{9}$$

Figure 2 shows the BC-induced error in these limiting cases (panel a) and for intermediate values of the information ratio (panel b) for an illustrative inversion with $n = 10$, $\varepsilon_C = 10$ ppb, and $\tau = 1.4$ h corresponding to $U = 5$ m s$^{-1}$ and $L = 25$ km. In the observation-limited case, the BC-induced errors decrease linearly with the distance from the upwind boundary. 175 The total BC-induced error is relatively small because of the strong constraint from the prior estimate. As the information ratio increases from $R \ll 1$, the BC-induced errors increase because of the decreased constraint from the prior estimate. The





errors increase the most in the most upwind grid cells. As the information ratio approaches one, the total error converges to the total flux needed to explain the error in the BC ($\tau^{-1}\varepsilon_C$). Further increases in the observational constraint decrease the length scale over which the BC bias influences the posterior solution but not the total BC-induced error integrated over the domain. In the observation-rich case, the BC-induced errors decay geometrically as the distance from the upwind boundary increases so that the BC-induced errors are limited to the most upwind grid cell.

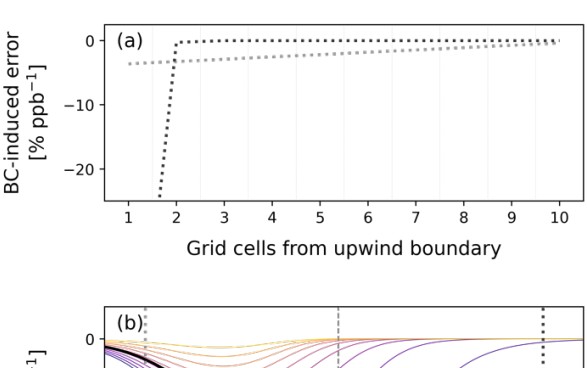

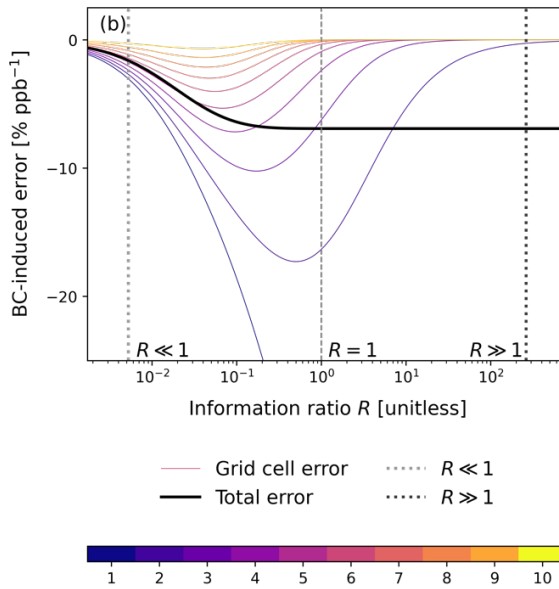

**Figure 2: BC-induced error on posterior surface fluxes as a function of the information ratio $R$ (Eq. (8)) for a one-dimensional atmospheric trace gas inversion over ten grid cells with constant parameters. The inversions assume $\tau = 1.4$ h corresponding to $U = 5$ m s$^{-1}$ and $L = 25$. The solid lines (b) represent the error induced by a 10 ppb BC bias relative to the prior emissions of 25 ppb h$^{-1}$ in each grid cell (thin lines) and the total error integrated across the full domain (thick black line). The dashed line corresponds to an information ratio of 1, while the dotted lines show representative small and large values for which the limiting cases described in Eq. (9) hold. The distribution of the BC-induced errors across the domain for these representative values is also shown (a).**



We also compare the boundary and buffer methods for correcting BC biases within the framework of the constant parameter, one-dimensional inverse model. We consider for this purpose an inversion of a single grid cell. For the boundary method, we use a BC prior error standard deviation equal to the constant BC bias $\varepsilon_C$ and update the Jacobian matrix to include the sensitivity of the grid cell average observation to the BC. For the buffer approach, we add a buffer grid cell and scale its prior error standard deviation by a factor of $p$.. We assume observations are equally distributed over both grid cells. We solve both inversions. The posterior fluxes are equivalent when

$$p = \frac{\varepsilon_C}{\tau \sigma_A \sqrt{R + 2}}. \tag{10}$$

Assuming steady state and knowledge of the domain wind speed, the buffer scale factor $p$ can be chosen so that buffer method performs equivalently to the boundary method. The scale factor goes to infinity for very small values of the information ratio (observation limited) and to zero for very large values (observation rich). The dependence of the scale factor on residence time implies that the buffer approach is more vulnerable to varying wind speeds than the boundary method.

### 2.3 Preview for boundary condition errors

Equation (5) can estimate the effect of BC-induced errors on the posterior solution once model transport is characterized by the construction of the Jacobian matrix. We estimate BC-induced errors to inform the choice of inversion domain before building the Jacobian matrix by applying Eq. (5) to a simple steady-state two-box model that generalizes a gridded flux inversion. Figure 1 (panels b and c) depicts the model. The domain has BC $c$ and constant wind speed $U$. All grid cells have representative grid cell length $L$ given the wind direction. We consider a grid cell (with index $j$) that is $j$ grid cells from the domain boundary and define the upwind grid cell (with index $j0$) as the aggregate of the upwind $j$ grid cells. We conduct an inversion of the average observations over these two grid cells $\boldsymbol{y} = [y_{j0} \quad y_j]^{\mathrm{T}}$ regularized by prior fluxes $\boldsymbol{x}_A = [x_{A,j0} \quad x_{A,j}]^{\mathrm{T}}$. We define the prior error covariance matrix as

$$\mathbf{S}_A = \begin{bmatrix} \sigma_{A,j0}^2 & 0 \\ 0 & \sigma_{A,j}^2 \end{bmatrix} \tag{11}$$

and use an equivalent structure for the observing system error covariance matrix. We assume the observing system error standard deviations account for the reduction in error resulting from averaging the observations. The Jacobian matrix is derived by steady-state mass balance (Supplement S1) to be a function of $\tau$ (Eq. (7)) given by



$$\mathbf{K} = \tau \begin{bmatrix} j & 0 \\ j & 1 \end{bmatrix}. \tag{12}$$

We solve for the effect of a constant BC bias $\boldsymbol{\varepsilon}_C = [\varepsilon_C \quad \varepsilon_C]^T$ on the posterior flux in the grid cell of interest $\Delta\hat{x}_j$ by applying the diagnostic (Eq. (5)) to compute $\Delta\hat{\boldsymbol{x}} = [\Delta\hat{x}_{j-1} \quad \Delta\hat{x}_j]^T$. This defines the preview

$$\Delta\hat{x}_j = -\frac{(\tau^{-1}\varepsilon_C)R_j}{1 + \beta R_j + R_{j0} + R_j R_{j0}} \tag{13}$$

where

$$\beta = \frac{j^2 \sigma_{A,j0}^2}{\sigma_{A,j}^2} + 1 \tag{14}$$

and $R_{j0}$ and $R_j$ refer to the information ratio (Eq. (8)) in the upwind and selected grid cell, respectively, with $\tau_{j0} = j\tau$. For grid cells abutting the boundary, all upwind values are set to 0 corresponding to an inversion solved only for the grid cell of interest excluding any upwind components. The preview estimates the BC-induced error for each grid cell in the domain by combining the effect of prior flux variability and observation density (as represented by the difference in the information ratio and prior flux uncertainty between the upwind and selected grid cell) with the decay in errors as distance from the domain edge increases (as represented by an inverse quadratic in $j$). The preview estimate also depends inversely on residence time.

By analogy to our one-dimensional model, we consider the observation-limited and observation-rich limiting cases assuming constant prior fluxes, prior flux standard deviations, observation density, and observing system errors. In this case, $\sigma_{A,j0}^2 = j^2 \sigma_{A,j}^2$ and $\sigma_{O,j0}^2 = \sigma_{O,j}^2/j$ assuming the observing system errors are uncorrelated. The resulting BC-induced errors

$$\Delta\hat{x}_j = \begin{cases} -(\tau^{-1}\Delta c)R, & R \ll 1 \\ -(\tau^{-1}\Delta c) \, j^{-5}R^{-1}, & R \gg 1 \end{cases} \tag{15}$$

match the general form of the one-dimensional approximation (Eq. (9)), but with the geometric decay in the observation-controlled limiting case approximated as quintic decay, which results from the dependence of the upstream prior error standard deviation, observing system error standard deviation, and residence time on the grid cell index $j$.




**Table 1: Inversion parameters for simulation experiments**

| Inversion parameter | One-dimensional model | Two-dimensional model |
|---|---|---|
| Domain state vector dimension | 20 | 285 |
| Prior emission estimate | Randomly sampled from normal distribution with mean 25 ppb d$^{-1}$ and standard deviation 5 ppb d$^{-1}$ | Modified Express Extension of the Gridded EPA inventory for 2020[d] |
| Prior error standard deviation[a] | 50% | 50% |
| Prior boundary condition estimate | Boundary condition mean | Smoothed TROPOMI fields[e] |
| Boundary condition error standard deviation | 10 ppb | 10 ppb |
| Number of buffer grid cells | 1 | Clusters: 10 No clusters: 440 [f] |
| Buffer scale factor values[b] | Constant winds: 4.8 Varying winds: 6.8[c] | 1000 |
| Observing system error standard deviation[a] | 10 ppb | 15 ppb [g] |

[a] The error covariance matrices are assumed diagonal.

[b] The buffer scale factor is the factor $p$ applied to the prior error standard deviation in the buffer grid cell.

[c] We calculate the buffer scale factor with Eq. (10).

[d] We increase the Express Extension of the Gridded EPA inventory for 2020 (Maasakkers et al., 2023) by a factor of three so that the prior error statistics are consistent with the true emissions.

[e] We use the IMI default boundary conditions, which are given by monthly mean TROPOMI observations smoothed over ≈1000 km.

[f] We take as the inversion buffer zone the five concentric rings of 0.25° × 0.3125° grid cells around the outer edge of the domain. We
solve inversions using 10 buffer cluster elements and 440 native resolution buffer grid cells.

[g] The IMI assumes a constant error standard deviation of 15 ppb for all observations aggregated into errors for the averaged observations accounting for error correlations following Chen et al. (2023). We use the errors generated by the IMI directly.

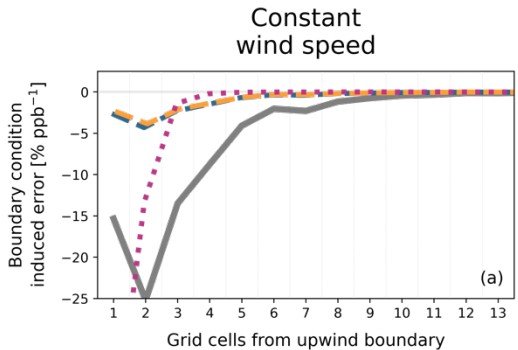
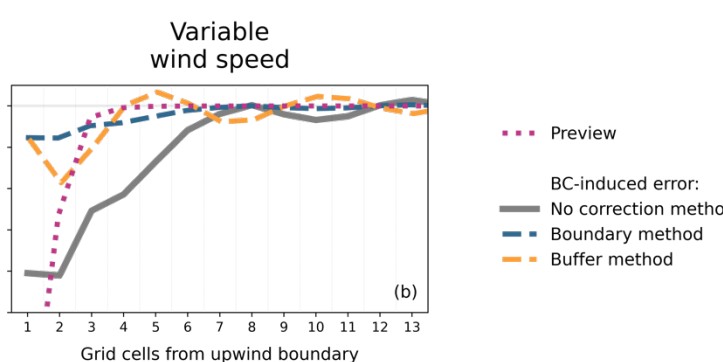

**Figure 3: Exact and predicted influence of a constant BC bias on the posterior fluxes generated by a one-dimensional inverse model using different BC correction methods. The lines represent the posterior error induced by a 10 ppb BC perturbation as a function of the number of grid cells from the upwind boundary using constant (a) and varying (b) wind speeds. The error is given as the difference between inversions solved with the true and biased BCs, normalized by the BC perturbation and the prior fluxes ($\Delta\hat{x}/(x_A\Delta c)$). The error is shown for inversions solved with no BC correction mechanism, the boundary method, and the buffer**
**method. The buffer grid cell is not shown. The panels also show the preview. Only the first 13 of 20 grid cells are shown; all errors approach zero after this point. The varying wind speed is a see-sawing wind with mean equal to the constant wind speed.**





## 3 One-dimensional numerical solution

We demonstrate the preview, diagnostic, and correction methods using illustrative inversions of a one-dimensional model for the horizontal transport of an inert trace gas (Sect. 2.2; Nesser, 2025). Table S1 summarizes the model parameters and Table 275 1 the inversion parameters. We use these demonstration inversions to illustrate the effect of constant and varying BC biases in inversions with constant and varying wind speeds (Sect. 3.1). We then generalize the results by varying the parameter choice to understand the variables controlling the behavior of the BC-induced errors (Sect. 3.2).

### 3.1 Constant and variable boundary condition perturbations

We quantify the sensitivity of posterior fluxes to BC biases by perturbing the true BC in our one-dimensional model, solving 280 the inversion, and comparing the posterior fluxes to those generated by an inversion solved with the true BC (Nesser, 2025). The inversions apply no other sources of bias (including transport errors) to isolate the effect of BC biases. The domain-average information ratio $R = 0.2$ for these demonstration inversions reflects the low DOFS achieved (5 for inversions with constant wind speeds and 7 for varying wind speeds) despite the uniformity and large number of observations ($m = 1000$), consistent with real inversions (Varon et al., 2023a).

Figure 3 (panel a) shows the relative difference in the posterior fluxes induced by a constant 10 ppb BC perturbation for an inversion with constant wind speeds. Because the posterior fluxes and metrics respond approximately linearly to constant BC biases (Eqs. (5), (10), and (14)), we show results normalized for the perturbation. We also normalize by the prior fluxes, which are unaffected by BC biases. As expected from our theoretical analysis, the resulting error on average decreases as the 290 distance from the upwind boundary increases. Exceptions to the decreasing trend result from normalizing by the prior fluxes. The preview, which is calculated using the mean wind speed, captures the error decay. The diagnostic (not shown) perfectly quantifies the no correction method error as expected from Eq. (5) given the specification of the BC bias.

Figure 3 also shows the effect of the buffer and boundary methods. The boundary method corrects BC concentrations to 295 within 1 ppb of the true BC, while the buffer method changes the buffer grid cell emissions unphysically by a factor of -6.6. Both methods reduce the error induced by the constant boundary condition perturbation. Indeed, the methods perform almost identically as predicted by the steady state approximation (Eq. (10)). The uncorrected errors for the boundary method result from the decreased information available to correct the upwind fluxes (first two terms of Eq. (6)), while the residual errors for the buffer method result from uncorrected BC biases (third term of Eq. (6)). Importantly, BC-induced errors decay over a 300 similar number of grid cells regardless of buffer size so that multiple rows of buffer grid cells are preferable to large clusters. Combining the buffer and boundary methods produces results identical to the buffer approach.





The consistency of the metrics and of the correction methods results in part from the use of constant wind speeds, which is aligned with the assumptions used to derive the preview and the equivalence between the correction approaches. Figure 3

(panel b) shows the effect of varying wind speeds. The BC-induced bias consists of decaying upwind biases and structured downwind biases. The preview captures the upwind decay but not the downwind biases while the diagnostic (not shown) perfectly predicts both the upwind and downwind biases due to its representation of transport and perfect knowledge of the BC bias. Both correction methods decrease the upwind and downwind biases. The boundary method strongly reduces both the magnitude and the variability of the BC-induced errors while the buffer method decreases the magnitude but not the

variability of biases due to its dependence on wind speed (Eq. (10)).

Varying BC biases are unresolvable within inversions without prior knowledge of their structure. As a result, we consider the types of BC biases that are most important to avoid in an inversion. Because varying biases can be represented as the sum of periodic functions, we represent them with a series of periodic BCs with varying y-intercept, amplitude, and period number

(Supplement S3). As in the inversion with a constant BC perturbation and varying wind speeds, the BC-induced bias consists of decaying upwind biases and structured downwind biases. The magnitude of both the upwind and downwind errors depends on the mean BC bias and is largely independent of the bias frequency (Fig. S1), suggesting that reducing the mean BC bias is the most effective way of decreasing errors in the posterior fluxes. The boundary method removes this mean bias directly, reducing both upwind and downwind error. The buffer method reduces neither upwind nor downwind errors. These

results likely also apply to the common case of spatially variable BC biases for inversions optimizing gridded surface fluxes.

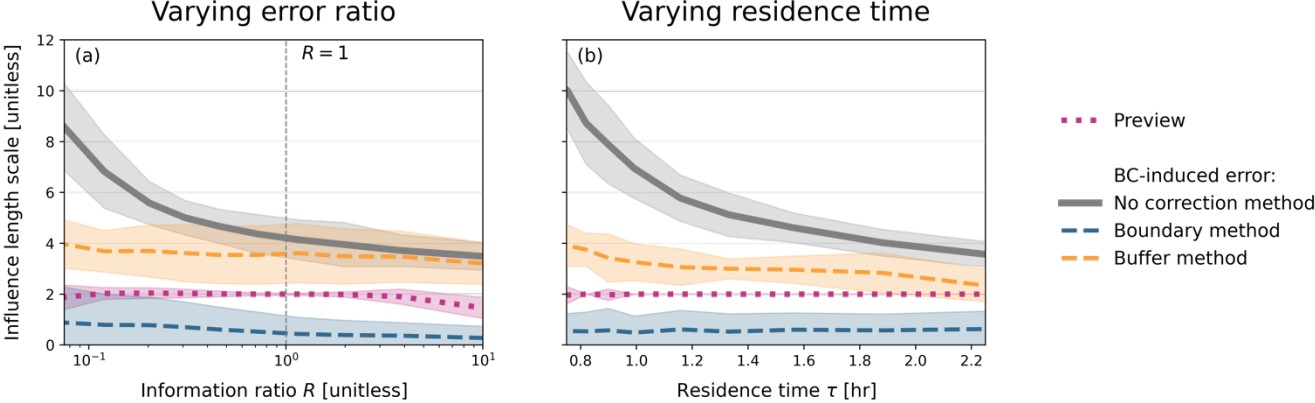

**Figure 4: Sensitivity of the BC-induced posterior error to changes in parameters in a one-dimensional inverse model. The vertical axis shows the influence length scale defined as the number of grid cells before which the prior-normalized posterior errors**

**resulting from a 10 ppb BC bias decrease below 0.25 ($\Delta \hat{x}/x_A < 0.25$). The lines show the influence length scale for the preview and for inversions solved with no BC correction method, the boundary method, and the buffer method. Shading gives the one standard deviation range for 100 inversions solved with different random prior fluxes. Panel (a) shows the influence length scale as a function of the information ratio $R$ (Eq. (8)) that describes the ratio of observing system error variances to the prior error variances in concentration units. The information ratio $R = 1$ is marked by a vertical dashed line. Panel (b) shows the influence**

**length scale as a function of grid cell residence time ($\tau$).**



## 3.2 Varying inversion parameters

BC-induced posterior biases are a function of the relative constraint from the observations compared to the prior fluxes as measured by the information ratio ($R$, Eq. (8)) and the grid cell residence time ($\tau$, Eq. (7)). To determine how changes in these quantities influence BC-induced biases, we define the influence length scale for our one-dimensional model as the

number of grid cells required for the BC-induced error normalized by the prior fluxes to decrease below 0.25 ($\Delta\hat{x}/x_A <$ 0.25). We vary the mean prior flux (from 20 to 50 ppb h$^{-1}$) and associated prior error standard deviations, observation density (from 50 to 200 observations per grid cell), and mean wind speed (from 3 to 10 m s$^{-1}$) and calculate the domain average information ratio and residence time for each inversion. Uncertainty is quantified as the standard deviation of the influence length scale across 100 inversions solved with random prior fluxes. All inversions use varying wind speeds and a

10 ppb BC perturbation, though the results are robust for constant wind speeds.

Figure 4 (panel a) shows the influence length scale as a function of the information ratio for an inversion with no correction method. As predicted from the one-dimensional and two-box models solved with constant parameters (Eq. (9), Eq. (15), and Fig. 2), the influence length scale decreases with the information ratio. The largest rate of change occurs in the observation-

limited regime as the observational constraint shifts the BC-induced errors into the upwind grid cells. In the observation-rich case, the influence length scale is relatively constant as a function of the information ratio. Figure 4 (panel b) also shows the inverse dependence of the influence length scale on grid cell residence time as predicted by Eqs. (9) and (15). As residence time increases, the influence length scale decreases because the relative contribution of the fluxes compared to the BC increases in the observed concentrations, improving the observational constraint.

Finally, we consider the performance of the correction methods and metrics. The boundary method outperforms the buffer cases in all cases, reducing the influence length scale to one or fewer grid cells. The buffer method reduces the influence length scale compared to the error associated with no correction mechanism in the observation-limited case but not the observation-rich case. The preview generates a stable estimate of the influence length scale that underestimates the true

influence length scale by about 50%, which is still adequate to inform the choice of inversion domain. The preview influence length scale is also relatively insensitive to variations in the information ratio and residence time because it uses variable grid cell information ratios and residence times instead of domain averages. As expected from the prescription of the BC perturbation, the diagnostic (not shown) perfectly predicts the influence length scale given by the no correction method error.

## 4 Two-dimensional solution

We extend the framework and understanding derived from the analytical solution and one-dimensional model to a two-dimensional demonstration inversion of TROPOMI-like methane column pseudo-observations over the Permian basin using the GEOS-Chem CTM as implemented by the Integrated Methane Inversion (IMI; Nesser, 2025; Estrada et al., 2024; Varon



et al., 2022). The Permian Basin is the largest oil producing region in North America and the subject of many inverse analyses (e.g., Zhang et al., 2020; Varon et al., 2023b; Vanselow et al., 2024). Table S1 summarizes the model parameters,

Table 1 gives the inversion parameters, and Fig. 5 shows the inversion domain, true emissions (panel a), prior emissions (panel b), and observation density as defined by real TROPOMI observations for May 2020 (panel c). The true BC is given by the IMI's TROPOMI-based smoothed BCs, which represents a best case scenario for BC variability (Estrada et al., 2024). Assuming a constant wind speed of 5 m s$^{-1}$ and using average values of the prior and observing system error standard deviations, the domain average information ratio is $R = 0.03$, reflecting the heterogeneous observational constraint.

We apply a spatially variable BC bias of 7.5 ppb, 10 ppb, 10 ppb, and 12.5 ppb to the northern, southern, eastern, and western boundaries using the Jacobian matrix for each edge of the domain as generated by the IMI. Figure 5 (panel d) shows the resulting percent error in the posterior emissions per ppb of mean BC bias normalized by the prior emissions. Unlike the BC-induced errors shown in the one-dimensional inversions, which used random prior fluxes, the distribution of the relative

errors is strongly a function of the prior error standard deviations. This dependence is consistent with the linear dependence of BC-induced errors on the information ratio, which is in turn proportional to the prior error variance, as predicted by the one-dimensional constant parameter inversion for $R \ll 1$ (Eq. (9)).

Figure 5 (panels f, g, and h) also shows the correction methods. For the buffer method, we test two sets of buffer grid cells

within the five concentric rings of grid cells around the outer edge of the domain. The first (clusters) aggregates the individual grid cells into 10 large buffer clusters using K-means clustering. The second (no clusters) uses the individual grid cells as buffers. In all cases, we scale the prior error standard deviation for the buffer elements by a factor of $p = 1000$. This very large, unphysical factor is chosen to compensate for the very small emissions around the edge of the Permian basin. The results mirror those found in the one-dimensional varying wind speed example. All correction methods decrease the BC-

induced errors. Despite relying on fewer than half of the observations used by the buffer method, the boundary method virtually eliminates BC-induced errors while avoiding unphysical flux corrections and significantly decreasing computational cost by reducing the domain size. The buffer method with no clusters outperforms the buffer method with clusters, suggesting that multiple rows of buffer grid cells are preferable to large clusters. Larger numbers of buffer grid cells may also better absorb biases with higher resolution spatial variability.

The preview calculated with a BC uncertainty of 10 ppb is shown in Fig. 5 (panel e). For the preview, we estimate the Jacobian matrix elements following Eq. (7) as implemented by Nesser et al. (2024) with a constant wind speed of 5 m s$^{-1}$, constant surface pressure of 1000 hPa, and a grid cell length scale given by the square root of each grid cell's area. We define the upwind length scale as the minimum distance from each grid cell's center to the border, dividing the domain into

395 four regions based on proximity to each boundary. Upwind emissions and observation counts are calculated as the cumulative sum of the median value of each row or column between each grid cell and its closest boundary. Observing



system errors are calculated by decreasing the single-observation uncertainty of 15 ppb by the square root of the observation count. The preview combines the decay in BC-induced error as distance from the domain boundary increases with information about the prior emission distribution and observation density. As a result, the preview captures the effect of the

BC bias nearest to the domain edge but is unable to predict the effect on emissions in the domain interior, consistent with the one-dimensional inversion with varying wind speeds. The diagnostic (not shown) perfectly captures the errors across the full domain as expected from the specification of the BC uncertainty.

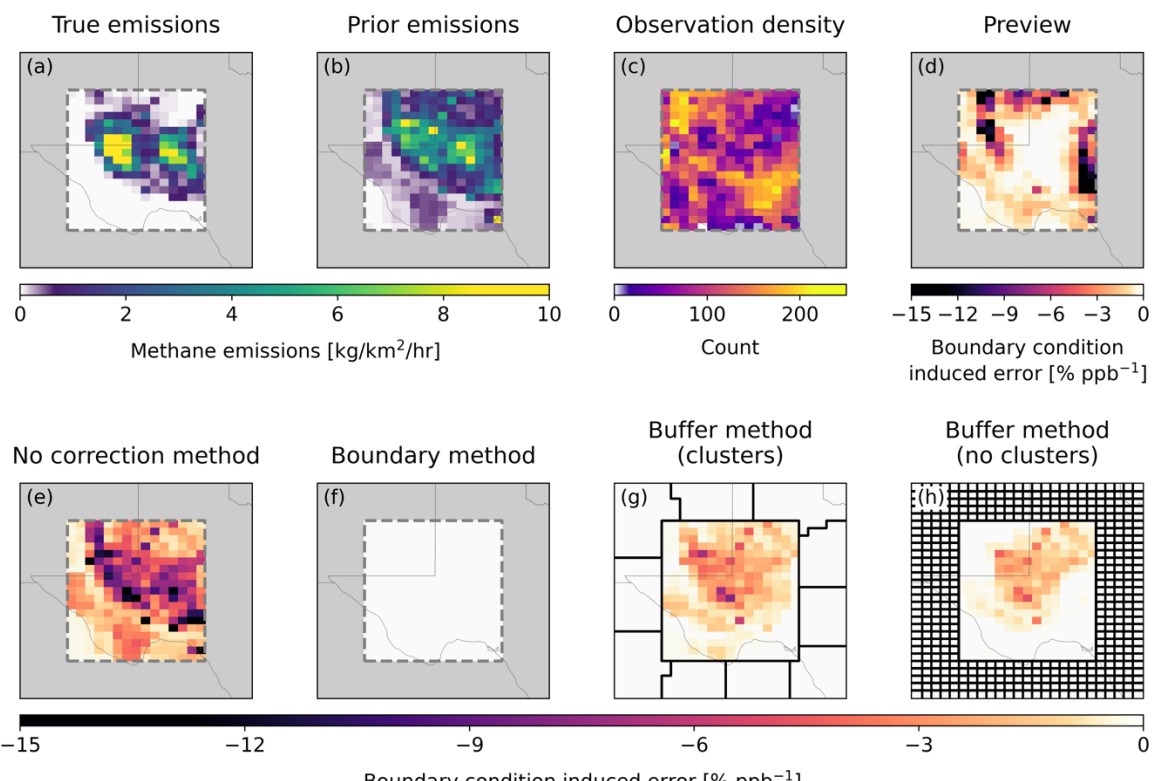

**Figure 5: Exact and predicted influence of a constant BC bias on a two-dimensional simulation inversion over the Permian basin in Texas using different BC correction methods. True emissions from the Environmental Defense Fund high-resolution inventory (a; Zhang et al., 2020) and true BCs from smoothed TROPOMI observations are used to generate pseudo-observations over the Permian basin (dashed line) with observational density (c) given by the TROPOMI methane observations for May 2020. The inversion is regularized by prior emissions given by the Express Extension of the Gridded EPA Inventory for 2020 multiplied by a**

**factor of three so that the prior error statistics include the true emissions (b; Maasakkers et al., 2023). The error induced by a spatially variable BC bias with mean 10 ppb is calculated as the difference between inversions solved with the true and biased BCs normalized by the prior fluxes and the perturbation to yield the approximate percent error induced per ppb of BC bias. The error is shown for inversions solved with no correction method (e), the boundary method (f), and the buffer method using ten buffer clusters (g) and five rows of native resolution grid cells (h). The dark black outlines show the buffer clusters or grid cells where the**

**prior uncertainties are artificially inflated and for which posterior fluxes are calculated but not shown. The preview (d) is also shown.**



## 5 Conclusions

We developed and demonstrated a framework to predict and correct the effect of BC biases on the optimal (posterior) gridded surface fluxes generated by regional inversions of atmospheric trace gas observations using a chemical transport model (CTM). We proposed two metrics to predict BC-induced errors both before (preview) and after (diagnostic) the inversion is solved. The preview can inform the choice of inversion domain while the diagnostic improves posterior error quantification. We also considered two methods to correct BC biases as part of an inversion by optimizing the BC directly (boundary method) or by unphysically correcting grid cell fluxes outside the domain of interest (buffer method). Both methods can obtain identical error reductions in inversions with constant wind speeds, but the boundary method is more effective for inversions with variable wind speeds, provides a physical constraint, and reduces computational cost. Beyond the application to regional inversions of long-lived gases presented here, the framework is more generally applicable to high-resolution inversions of short-lived gases, the analysis of bias in the observations or CTM, and the treatment of initial conditions in global inversions.

We demonstrated our theoretical framework using a simple one-dimensional model for the horizontal transport of an inert trace gas, which represents the worst-case scenario for the propagation of BC biases to the posterior flux estimate. BC-induced errors on average decay with increasing distance from the domain edge while smaller downwind systematic biases result from variability in the CTM wind speeds or BC biases. The length scale over which BC-induced errors have a significant effect on the posterior fluxes is minimized when the observations provide a strong constraint across the domain, which limits the bias to the most upstream grid cells. The boundary method significantly reduces the influence length scale and the magnitude of downwind errors in all cases, while the buffer method decreases the influence length scale only for observation-limited inversions. The preview generates a stable estimate of the influence length scale, while the diagnostic perfectly predicts the BC-induced error when the BC perturbation is specified. The results are robust for constant and variable BC biases.

We extended the framework to a two-dimensional demonstration inversion over the Permian Basin in Texas using model transport from the GEOS-Chem CTM. The boundary method functionally eliminates the BC-induced errors at much lower computational cost despite relying on fewer than half of the observations used in the buffer method. The buffer method decreases but does not eliminate BC-induced errors, with performance improving as the number of buffer clusters increases. The preview combines the expected decay in BC-induced errors as distance from the domain boundary increases with information about the prior flux distribution and observation density. As a result, the preview accurately predicts the BC-induced errors closest to the domain boundary. The diagnostic accurately describes the errors across the full domain.



## Code availability

The code and data for both the one-dimensional and two-dimensional simulation experiments are available at
450 https://doi.org/10.5281/zenodo.15742048 (Nesser, 2025). The IMI v2.0 used to generate the inputs for the two-dimensional example is available at https://github.com/geoschem/integrated_methane_inversion/releases/tag/imi-2.0.1.

## Author contributions

HN and DJJ designed the study. HN, DJJ, and KWB contributed to the theoretical development. HN conducted the theoretical analysis and demonstration applications. MDT contributed to the proof of Eq. (9). KWB, DJV, CR, AT, FJCS,
ER, and JDM discussed the results. HN, DJJ, and KWB wrote the paper with input from all authors.

## Acknowledgments

HN was funded in part by an appointment to the NASA Postdoctoral Program at the Jet Propulsion Laboratory, California Institute of Technology, administered by Oak Ridge Associated Universities under contract with NASA. The research was carried out at the Jet Propulsion Laboratory, California Institute of Technology, under a contract with the National
Aeronautics and Space Administration (80NM0018D0004). This work was funded in part by ExxonMobil Technology and Engineering Company. We thank Kimberly Mueller for her feedback.

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
