# Peer review of "Predicting and correcting the influence of boundary conditions in regional inverse analyses"

_EGUsphere, 2025_

## Author Comment (AC1)

**Reviewer 1**

This manuscript is well-prepared, well-organized, and well-written. It is also a very useful study for practitioners of atmospheric inversions at regional and local spatial scales, where background errors are complex and often large in magnitude, because decreasing spatial scale usually increases variability (and error) in the background estimate. Analysis at these scales is becoming more and more common, as researchers seek to provide emissions data at "actionable" scales, i.e. for a given oil and gas basin, city, or state. This should be published with minor edits.

Most of my comments below are requesting clarifications of legends/figures and of some more intuitive explanation of one of the results -- although I believe part of my confusion was due to a mislabeled figure.

We thank the reviewer for her kind words and for the time she took to provide detailed feedback.

Detailed (minor) comments:

L16 "a gridded flux estimates" should be made either singular or plural Thanks for catching this; we've changed it accordingly.

L196 two periods after p

Thanks for catching this; we've changed it accordingly.

Fig S1: Having trouble understanding the lines in the lower three panels relative to the legend - both the "no correction method" and the "downwind error" are in a similar shade of gray, but I do not see two different lines in the plots. Also what is the light blue dashed line (e.g., around 0.2 in panel (d))? I wonder if the color palette could also be changed to show larger differences between the lines, while still making them color-blind friendly?

What about the boundary method, doesn't it reduce upwind differently from downwind? Please see our combined response to this and the next point below.

L317: Related to the above is the understanding of the text here - I am not seeing the boundary method reducing both upwind and downwind error in panel S1(d), I only see one line there. Also the buffer method does seem to reduce downwind errors relative to the "no correction method" case, I think? I think I am just not understanding the legend, so please clarify.

Thanks for these suggestions. We remade Figure S1 to improve the clarity by mirroring the structure of Figure 3. We also added a similar summary figure into the main text and separated the discussion on periodic perturbations into their own section. Hopefully these changes improve clarity.

To answer your question: the boundary and buffer methods decrease the upwind error because the upwind error is strongly a function of the mean bias in the BC. As for downwind errors: You are correct that the boundary method isn't able to correct these errors unless we optimize more boundary elements. We've now corrected this description in the text.

Fig. 4 - perhaps title of the left panel could be changed to "information ratio" for consistency. Thanks for the suggestion. We've changed this.

L352 should say "outperforms the buffer method in all cases"

That's correct, but we have deleted this sentence after remaking Figure 4 with a few minor fixes to our model implementation.

L355 -- is this likely to be the case for other inversions? (that the preview underestimates the influence length scale with no correction by  $\sim$ 50% but overestimates it for the boundary adjustment by  $\sim$ 50%). I imagine that would be informative for people who would like to use it.

We have improved the implementation of the preview so that the preview no longer has an obvious systematic bias compared to the true bias. The preview is now, if anything, an overestimate of the uncertainty, which makes it a useful estimate of the upper bound of BC-induced errors.

L364 - is the tower based non-IMI inversion included in these refs, perhaps it should be. Refs Varon et al 2023a and 2023b are the same

We added a citation to Barkley et al. (2023), which used tower sites to quantify Permian emissions. The Vanselow et al. (2024) citation is also a non-IMI citation. Thanks also for catching the repeated citation; we've corrected this.

L375 and Fig 5. I believe (d) and (e) are swapped when referred to in the text. This led to some confusion on my part but the overall question is why does the preview give such different spatial results from the real error? (i.e. why are these two panels so different)? Perhaps some intuitive explanation could be made here, and explanation with what regime we are in in terms of R. You're correct that (d) and (e) were swapped in the text. This has been corrected. We also have improved the implementation of the preview so that it now gives much better spatial agreement with the real error.

All the same, to clarify the role of the preview, we added text to explain the structure of the preview equation (Section 2.3). We write:

"As distance from the boundary increases, the BC-induced errors on average decrease due to the accumulation of both upwind fluxes and observations, which increases the upwind information ratio  $R_{j0}$ . Variability in the prior fluxes and observation density as quantified by the information ratio  $R_{j}$  is super-imposed on this decay."

L387 - please clarify - the boundary method reduces computational cost relative to the buffer method, but still increases it relative to no correction ,yes? How much more expensive is the buffer method in this example? How many elements are being estimated in the boundary method (one value per observation, or one value per edge)?

Thanks for the question! We added clarifying text on the dimension of the different inversions throughout Section 4. The standard inversion optimizes emissions in 285 grid cells, the buffer cases

use 10 additional clusters or 440 additional grid cells, and the boundary method uses 4 additional elements corresponding to the cardinal directions.

L395-400 It seems the preview-derived uncertainty spatial map is not correlated (or maybe it is negatively-correlated) with the map of the prior (and prior error). But the prior uncertainty does factor in through the beta term, can this be explained?

We have improved the implementation of the preview so that it now gives much better spatial agreement with the real error.

L425 Again clarify: reduces computational cost relative to the buffer method. Thanks for the comment. We've clarified this.

L445 – this was a useful summary of the difference in the preview vs. the full error, answering some of my questions on the earlier section, but I still think some reader confusion could be alleviated with an intuitive interpretation of the results in the previous section.

Thanks for the comment. We attempted to mirror some of the language in this summary in our revision of Section 4.

**Supplement notes:**

S1, line 8 add the definition of xj: "The area A converts the flux xj from units of mass per area...". Thanks for catching this! We now define  $x_j$ .

S2: derivation of Eq. 9. Says we will show that Eq. 10 is equal to delta xhat = -Gec in the extremes of R, this should read Eq. 9 I believe?

That's correct. We've changed this to Eq. 8. Note that the equations were renumbered due to the deletion of Eq. (6); the numbers here refer to the new numbering.

Fig S1 as noted earlier – I had trouble interpreting that figure due to the choice of colors and legend. Thanks for the comment. We separated the upwind and downwind results into separate panels and matched the color palette/line styles to the rest of the figures.

**Reviewer 2**

This work examines the role of boundary condition (BC) errors on regional inversions of long-lived trace gas emissions. As summarized in the conclusions, the findings of this work are preview (prediction) and diagnostic metrics of BC induced errors in the posterior fluxes, and the comparison of two inversion methods for treating BC errors. These are derived and examined through 1D, two-box, and 2D examples. While the aims of this work are laudable, overall, much of it struggles to demonstrate relevance, accuracy, or practical usefulness to the problem at hand. In particular, the applicability of the 1D and 2 box results, and the preview estimates is not clear. Several of the 1D results don't translate to the findings of the 2D case, in terms of the dependence of the BC error on length from the boundary for any method, or the accuracy of the preview estimate. The preview

estimate seems to miss key functional relationships and lacks applicability as a bounding estimate in the 1d and 2-box model; in the 2D application it simply seems inaccurate.

The diagnostic of BC errors is also presented as a conclusion. However, this diagnostic (equation 5) is not a new result (it is a straight forward application of theory from Rodgers). It is also rather useless in practice. While it does allow for correct quantification of the BC induced error on the posterior emissions, it requires precise knowledge of the boundary condition error (epsilon\_c) to begin with. In practice, if one does know the BC error to begin with, one corrects the BC's (or the obs) by this amount before doing the inversion. More useful diagnostics would be knowing how errors in an estimate of the BC bias may impact the solution, upper bounds on the impact of BC errors under particular conditions, etc.

Thus, the main useful findings of this work are related to the comparison of the buffer method (clustered or non-clustered) to the boundary method, and I suggest a revised manuscript focus much more on these. There are some outstanding questions related to their setup and findings on these aspects, see detailed comments below. The application / relevance of the 1D and two-box model results and preview equation need to be further demonstrated prior to their inclusion in a final manuscript. If omitting these aspects, a lot of my detailed comments below can be ignored.

We thank the reviewer for taking the time they took to provide detailed and thoughtful feedback. We have responded to the comments in the above paragraphs in detail below, but to summarize:

- 1. **Preview:** We have improved the implementation of the preview. We now believe that it accurately predicts BC-induced errors and captures the key functional relationships. This is reflected in all figures.
- 2. **Diagnostic:** We agree that the diagnostic is a straightforward application of theory from Rodgers (indeed, we cite Rodgers exclusively in these paragraphs). We also agree that if a user has precise knowledge of the BC error, they would correct the BC directly before doing the inversion. In practice, we more often have knowledge of the BC *error statistics* (in the same way that we estimate transport errors) which can be used with Eq. (5) to estimate BC induced error, accounting for model transport. We now write,

"We define Eq. (5) as the diagnostic that estimates the effect of BC bias on the posterior fluxes given assumed BC error statistics and an explicitly constructed Jacobian matrix. BC error statistics can be estimated using methods similar to those used to estimate other sources of observing system errors (e.g., Heald et al., 2004)."

We also now use the periodic BC perturbation as an application of this statement: we calculate the diagnostic using the mean bias in the BC (which is an estimate of the error statistics) and show its ability to capture both the upwind bias and the structural components of the downwind bias attributable to transport.

- 3. **Correction methods:** We have clarified the implementation of the buffer and boundary methods. These remain a major focus of the manuscript.
- 4. **Simple models:** A major argument of this work is that simple models can be used to understand complex systems. We hope that this is much clearer now that we have improved the implementation of the preview. We also agree that simple models have their limitations, and would argue that showing these limitations is important to the description. We now outline this more explicitly, stating in the conclusion:

"We extended the framework to a two-dimensional demonstration inversion over the Permian Basin in Texas using model transport from the GEOS-Chem transport model. The BC-induced biases do not decay with distance from the boundary but instead correlate with the prior flux estimate due to the concentration of emissions in the center of the domain. Despite the difference in the distribution of BC-induced biases compared to the one-dimensional model, we find similar performance of the correction methods and metrics."

**Comments:**

That the preview estimates generate a "stable estimate of the influence length scale" is an overly generous assessment. In Fig 3, the preview underestimates the length scale over which BC induced error is significant by several a factor of two or more, hitting 0 at 3-4 grid cells while the actual error gets to zero around 7-10 grid cells.

Thanks for your comment. We've improved our implementation of the preview and revised the text accordingly, but we removed the description of the preview as a stable estimate.

In Fig 4, the preview completely misses the functional dependence on R or tau, and it unfortunately also underestimate the length scale and as such does not provide a useful upper bound. In the 2D tests of Fig 5 / section 5, the preview of the boundary condition induced error (Fig 5d) looks nothing like the actual boundary condition induced error (Fig 5e). Line 445 claims the preview estimate is accurate near the boundary, but this isn't rigorously shown, nor is it discussed when / why the preview differs substantially from the actual error.

Thank you for your comment. We will respond to your comments on Figure 5 separately.

Figure 4: We have improved our implementation of the preview and it now captures the functional dependence on R and tau. It also now marginally overestimates the BC-induced error so that it represents an upper bound.

Figure 5: We've improved our implementation of the preview and now believe that the preview strongly resembles the true BC-induced error. We also now quantify the agreement, stating:

"The preview models the decay in BC-induced error as distance from the domain boundary increases, accounting for the prior emission distribution and observation density (Sect. 2.3). As a result, the preview most accurately captures the effect of the BC bias nearest to the domain edge with a Pearson correlation coefficient of r = 0.76 between the BC-induced biases and the preview for grid cells within three rows of the boundary. The performance of the preview degrades near the center of the domain, though there is still good agreement between the BC-induced biases and the preview for the full domain (r = 0.67). The preview is larger than the BC-induced bias in most (63%) of the grid cells so that the preview represents an error upper-bound."

Thus, what is the value of the preview? Is it a bound in any way? Does it inform an inversion setup in any way?

We hope that our responses to the previous points address this concern. We believe that with the improved implementation, it models an upper bound on BC-induced biases.

The 1D analysis in sections 2 and 3 imply there is a distance from the edge at which point the boundary condition error becomes minimal, usually around 8-10 grid cells. The preview estimate again estimates this to be the case. But the actual error shown in e is spread throughout the domain. Thus I'm not sure what is the value of the analysis in sections 2 and 3, which seemed to imply that there was a "safe" distance from the edge even without any boundary or buffer method applied. Thank you for your comment. We would make two points here:

- 1. The analysis in sections 2 and 3 shows that downwind biases induced by variable wind speeds, variable BC biases, or both, can be systematic and significant so that there is no "safe" distance from the edge without applying a correction method. We now include a separate section on variable BC biases to emphasize the importance of downwind biases.
- 2. The 2D model shows an important exception to the general pattern of decay for BC-induced biases in the common case where the inversion domain has large fluxes in the center of the domain and low emissions around the edges. We've now clarified that this is an important exception both in Section 4 and the Conclusions, writing:

"We extended the framework to a two-dimensional demonstration inversion over the Permian Basin in Texas using model transport from the GEOS-Chem transport model. The BC-induced biases do not decay with distance from the boundary but instead correlate with the prior flux estimate due to the concentration of emissions in the center of the domain. Despite the difference in the distribution of BC-induced biases compared to the one-dimensional model, we find similar performance of the correction methods and metrics."

The error in 5e also shows a lot of structure connected to the prior, rather than the boundary conditions. I'm thus skeptical about it being "boundary condition induced error."

Thank you for your comment. The only thing that changes in our inversions is the BC, so this is BC-induced error. This pattern occurs because there are very small emissions outside of the center of the domain so that the inversion is unable to successfully absorb BC biases in these grid cells. We've added this explanation to the text:

"Unlike the BC-induced errors shown in the one-dimensional inversions, which used random prior fluxes, the distribution of the relative errors is strongly a function of the prior emissions. This likely results from the skewed distribution of the emissions across the domain, which limits the inversion's ability to correct fluxes near the domain edge."

The setup also seems to violate a key assumption of the methods, in that x\_A doesn't not seem to be a sample of an emissions distribution with a mean of x\_T. The true emissions x\_T are zero in large parts of the domain. A random sample around this with standard deviation of sigma\_A would lead to negative prior emissions. However, there are no negative prior emissions. It seems they've taken the true inventory and the prior inventory from independent sources. There's an inflation to the prior to make some statistics match (Table 1, comment d), but this is rather vague and I'm rather skeptical that it makes x\_A a sample of a gaussian distribution with mean x\_T and variance s\_A.

Thanks for your comment. In theory, we agree that Bayesian inversions assume that the prior is unbiased compared to the truth (and normally distributed). We would expect for the true emissions to be positive over a region like the Permian, so it is theoretically justified for the prior (which represents the mean of the prior distribution) to always be positive over the region. Our prior uncertainties do allow for negative emissions and negative posterior emissions are therefore possible.

In practice, the truth is unknown and the prior serves as the "best first guess" to regularize the undetermined solution. The prior is normally informed by the best-available data that may or may not reflect the "true emissions." We believe our choice of true emissions and prior is therefore representative of the true uncertainty that is present in real inversions.

2D Boundary method: Optimize how many separate elements along the boundary? One for each horizontal and vertical grid cell? In that sense, how is this different than a setup with a buffer method, with a buffer that is one grid cell thick? The only hint I see here is line 104 that says "one or more." Given this is the best approach being put forward, some more details could be provided. Thanks for your question. We now explicitly state that we optimize one element for each edge of the boundary: "For the boundary method, we optimize a mean bias correction along the northern, southern, eastern, and western boundaries."

General: I'm a bit confused about what model is being used for which test / set of results. Let me see if I have this correct. Section 2 presents a 1D (Section 2.2) and then two-box (Section 2.3)

model. Fig 2 shows results from the 1D model. Fig 3 also presents results from the 1D model, but described in section 3. Why one of these is in the methods and the other the results is a bit confusing. Thank you for your comment. We don't define the sections as "Methods" and "Results" but instead as "Quantifying the effect of boundary condition errors," "One-dimensional numerical solution," and "Two-dimensional solution." We renamed the first section to "Analytical solution for the effect of boundary condition errors" and the last to "Two-dimensional numerical solution" to improve clarity.

As you describe, Section 2 presents the theoretical basis focused on the analytical description of BC biases. Section 2.1 describes the analytical framework. Section 2.2 describes a one-dimensional model with constant inversion parameters. Section 2.3 generalizes this one-dimensional model into two dimensions by (a) reducing the dimension to a two-box model and (b) by allowing variability in inversion parameters. We describe this in the introduction to Section 2, which we've edited for clarity:

"We describe the analytical inverse solution, which we use to derive an exact solution (diagnostic) for the sensitivity of the posterior fluxes to a BC bias (Sect. 2.1). We apply this diagnostic to a simple one-dimensional transport model to determine how these errors depend on constant inversion parameters (Sect. 2.2). We then generalize the results by using a two-box model applicable to two-dimensional inversions with variable inversion parameters to derive the preview metric that predicts the sensitivity of the posterior fluxes to BC biases before conducting the inversion to support domain specification (Sect. 2.3)."

Similarly, as you describe, Section 3 presents applications of the theoretical basis using a numerical one dimensional-model similar to that described in Section 2.2, but with non-uniform inversion parameters. Whereas section 2 focuses on the analytical formulation and requires large simplifications in the inversion assumptions, this is a numerical implementation that allows for the introduction of variability in prior uncertainties, wind speeds, and boundary condition biases. We describe this in the introduction to Section 3, which we've also edited for clarity:

"We demonstrate the preview, diagnostic, and correction methods using illustrative, numerical inversions of a one-dimensional model for the horizontal transport of an inert trace gas (Sect. 2.2; Nesser, 2025)."

Also, the text on Fig 3 (line 287) refers to Eq 14, which is for the 2 box model. Also I think they didn't mean to refer to Eq 14 at all here, since it doesn't contain epsilon\_c — perhaps they meant Eq 13?

Thank you for your comment. We've deleted this text.

Fig 4 caption says it is based on the 1D model, so I would think the "preview" line corresponds to Eq 9. However the text (lines 343, 347) refers to Eq 9 and Eq 15, and Eq 15 is for the two-box model, not the 1D model. How is the 2 box model used for the results shown in Fig 4?

Thank you for your comment. The preview always corresponds to Eq. (12). We added references to Eq. (12) to mentions of the preview throughout the text and did the same for the diagnostic. We also simplified the references in lines 343, 347 (now lines 513, 517) to only refer to Eq. (8), which gives the limiting cases for the one-dimensional model with constant parameters. Note that the equations were renumbered due to the deletion of Eq. (6); the numbers here refer to the new numbering.

Fig 4: I'm having trouble thinking about how these results are applicable, given that the metric is grid cell number (which seems arbitrary). Can this instead be presented in terms of a length scale related to wind speed, grid size, and chemical lifetime? The latter would be much more useful and general. Otherwise, this plot tells me I can get by with a buffer of 4-8 grid cells, regardless of whether the grid cells are 50 m and I'm inverting for CO2 fluxes under windy conditions, or the grid cells are 500 km and I'm inverting for NOx fluxes under stagnant conditions, which doesn't seem correct.

Thanks for your question. We would argue that grid cell number is the relevant quantity for two reasons. First, we find that the pattern of BC-induced error decay is surprisingly invariant to inversion configuration. Changes to the correction methods (including buffer size) alters the magnitude of this pattern but not the distribution. This suggests that inversions gain information about the contribution of the BC to the observed concentrations from the modeled gradient over multiple grid cells. Second, the computational cost of analytical inversions scales as a function of the number of optimized grid cells, so that the influence length scale provides a computational constraint. We've clarified this in the text:

"To determine how changes in these quantities influence BC-induced biases, we define the influence length scale for our one-dimensional model as the number of grid cells required for the BC-induced error normalized by the prior fluxes to decrease below 0.25 ( $\Delta \hat{x}/x_A < 0.25$ ). The influence length scale captures the strong dependence of the BC-induced errors on the number of grid cells from the domain boundary and reflects the additional computational cost associated with optimizing these grid cells."

427: The claim that this applies to short-lived gases seems to be an overstretch. As alluded to in comments above, I doubt the direct applicability of some of the assumptions and equations developed here for short-lived gases, for whom chemical lifetime needs to be accounted for somewhere, and also given that atmospheric chemistry often leads to nonlinearities in short-lived species that are not accounted for in the present framework.

Thank you for a comment. We've deleted this clause. For the purposes of clarification, we were referring to limited-domain inversions of short-lived trace gases, for which non-linearities can often be neglected (depending on the comparative lifetime of loss due to transport vs. chemistry).

26: Does this only apply to long-lived trace gases? It's not an issue for e.g. NH3, NO2, etc. Also, regional inversions of trace can concentrations don't necessarily employ a CTM. They may use online models, or Lagrangian back trajectory models, plume models,...

Thank you for your comment. We now specify that we are referring to long-lived trace gases. We've also removed any reference to CTMs and instead just reference transport models, which will includes inversions done with LPDMs or plume models. Our framework is generalizable to any Bayesian system, whether the Jacobian is constructed using an Eulerian or Lagrangian model, so we are grateful for the opportunity to improve our clarity here.

70: I don't agree that a long spin-up makes initial conditions unbiased compared to the BCs. If the BCs are biased owing to transport, but initial conditions are biased owing to incorrect fluxes within the domain, these biases could be different.

Thank you for your comment. We would argue that the question of whether the ICs are unbiased with respect to the BCs is independent of questions of transport or flux bias. In the example of an Eulerian model, it is a question of whether the gridded concentrations used as initial conditions are consistent with the concentrations defined as boundary conditions. We hope this clarifies what we mean. We have altered the text to state: "We assume as is standard that the transport model initial conditions are given by a sufficiently long spin-up simulation driven by the BCs so that the initial conditions are consistent with (or, equivalently, unbiased to) the BCs."

71: c would include contributions from the ICs as well. These could be different than those from the BCs, see above. If one wants to avoid this, one cold assume the inversion period is sufficiently long enough that the contribution from ICs becomes negligible compared to BCs and fluxes within the domain.

Thank you for your comment. We are assuming the ICs are given by the transport of the BCs into the domain, consistent with your explanation. We assume that the spin-up period (not the inversion period) is "sufficiently long enough that the contribution from ICs becomes negligible compared to BCs and fluxes within the domain." We changed the text to state: "We assume as is standard that the transport model initial conditions are given by a sufficiently long spin-up simulation driven by the BCs so that the initial conditions are consistent with (or, equivalently, unbiased to) the BCs."

85: This might be a bit clearer if defining epsilon directly as epsilon =  $y - (K x_T + c)$ Thank you for your suggestion. We now define epsilon in this way.

Eq 6: The last term does not seem correct to me, or there is an inconsistency between the text describing the setup and the equations presented. Perhaps not a big deal as this equation never is actually used, as far as I can tell? But it should at least be correct if presented. My derivation is as follows.

Retrieval with true boundary conditions and an observing system error eps\_T that is owing only to things like obs error, representational error, model error, etc., is (focusing only on the G eps terms for simplicity):

$$eps_T = y - (K x_T + c_T)$$

 $x_1 = x_A + ... + G eps_T$

Retrieval with boundary correction method and boundary error eps\_c, omitting augmented state vector elements:

eps' = y -
$$(K x_T + c_T + eps_c)$$
 = eps\_T + eps\_C
 $x_2 = x_A + ... + G'$  eps'
= x A +... + G' (eps\_T + eps\_C)

Taking the difference:

$$x_2 - x_1 = ... G' eps_T + G' eps_C - G eps_T = ... (G' - G) eps_T + G' eps_C$$

This differs from Eq 6, even if you assumed  $eps_T = 0$ .

Thank you for your comment and for taking the time to check our math. You are correct that the result here is incorrect. However, we decided to remove this equation since it doesn't contribute materially to our analysis.

Eq 15: Should this be epsilon\_c rather than  $\Delta c$ ? Otherwise, define  $\Delta c$ ? Thanks for catching this. This is correct, and we've changed it accordingly.

Eq 15: The R << 1 case is clear. It's not clear to me how you are getting the R >> 1 limit from 13. It seems like it should just be -  $(tau^-1 eps_c)/R$ , since for limiting cases Rj is approximately equal to R0 (?), and thus the denominator reduces to  $1 + (j^4 + 1)R + R + R^2$ , which would be dominated by R^2 for large R. Even including the j^4 R in the denominator doesn't get me to the expression provided in Eq 15.

Thank you for your question. In fact,  $R_{j0} = j^4 R_j$ . Note that this is a slight correction to our previous analysis, which assumed  $R_{j0} = j^5 R_j$  due to incorrect scaling of the prior flux uncertainty (which we assume are uncorrelated). We now describe this simple derivation in the text.

"By analogy to our one-dimensional model, we consider the observation-limited and observation-rich limiting cases assuming constant prior fluxes, prior flux standard deviations, observation density, and observing system errors. In this case,  $R_{j0} = j^4 R_j$  due to the dependence on the number of grid cells j of the upstream observation count  $(m_{g,j0} = jm_g)$ , grid cell residence time  $(\tau_{j0} = j\tau)$ , and prior flux uncertainty (assuming the uncertainties are uncorrelated,  $\sigma_{A,j0} = \sigma_A/\sqrt{j}$ ). The resulting BC-induced errors

$$\Delta \hat{x}_j = \begin{cases} -(\tau^{-1} \varepsilon_{\mathcal{C}}) R, & R \ll 1 \\ -(\tau^{-1} \varepsilon_{\mathcal{C}}) j^{-4} R^{-1}, & R \gg 1 \end{cases}$$
 (14)

match the general form of the one-dimensional approximation (Eq. (8)), but with the geometric decay in the observation-controlled limiting case approximated as quartic decay, which results from the dependence of the upwind prior error standard deviation, observing system error standard deviation, and residence time on the grid cell index *j*."

284: You mean consistent with a specific real inversion? There are many inversion systems beyond Varon 2023a, for different observational datasets, trace-gases, resolutions, and transport models. It's not clear to me that R=0.2 is common across all such inversions.

Thanks for your comment. Figure 4 in Varon et al. (2023) shows the Degrees of Freedom for Signal (DOFS) as a function of the number of observations for a series of weekly inversions over the Permian basin, which shows that there are approximately 1.4 DOFS per 1000 TROPOMI observations. We agree that there are many inversion systems with many different inversion parameters and different regions, but we would argue (1) that the Permian is close to ideal for the DOFS per observation we would expect and (2) that differences in other parameters are not likely to substantially change the order of magnitude of the DOFS per observation. We therefore use Varon et al. (2023) as evidence for the consistency of low information content in our inversion despite the large number of observations.

387: Outperforms in terms of error, but also probably is more expensive, in terms of computing K? So there's a tradeoff here.

Thanks for the question! We added clarifying text on the dimension of the different inversions throughout Section 4. The standard inversion optimizes emissions in 285 grid cells, the buffer cases use 10 additional clusters or 440 additional grid cells, and the boundary method uses 4 additional elements corresponding to the cardinal directions.

382: It seems like p should change with the number of buffer grid cells. With only 10 buffer clusters, it should be much larger, compared to the gridded buffer?

We find that the error reduction as a function of p asymptotes quickly. In this case, we find that 1000 is adequate in all cases to minimize the induced error. We now specify this in the text:

In the 1D case: "Increasing p improves performance, but the error reduction quickly asymptotes..."

In the 2D case: "In all cases, we scale the prior error standard deviation for the buffer elements by a factor of p = 1000, which we find is sufficient to minimize BC-induced errors in all tested inversions.

**Minor edits:**

16: estimates —> estimate We corrected this.

22: "grid cell fluxes outside the main of interest" isn't very clear — for a regional inversion, wouldn't the domain of interest be the entire region, and anything outside the BCs? I suppose this distinction will become clear upon reading the paper, but at present for a reader skimming the abstract alone, it could be clearer. The abstract is far from lengthy.

Thank you for your suggestion. We now write: "We compare two methods to correct BC biases as part of an inversion, either directly by optimizing BC concentrations (boundary method) or indirectly by expanding the domain and correcting grid cell fluxes outside the region of interest (buffer method)."

27: "background concentrations given by boundary conditions" is a bit of an odd expression. Perhaps more direct to say "assuming boundary conditions"

Thanks for the comment. We changed this phrasing.

30: What is a "boundary condition concentration"? I guess this is just a bit of imprecision in the language used here. I'd assume a boundary condition is a lateral flux into the domain. This depends upon some assumed concentrations at the boundary, which would be "boundary concentrations", not boundary condition concentrations? BC biases could be owing to the transport as well as the boundary concentration, yes? Some care with the wording regarding this throughout could be useful. For example, line 34 would be "Boundary concentrations used to define BCs can be provided by..."

Thanks for your suggestion. By "BC concentration" we mean the concentration fields used as boundary conditions. We now write simply "BCs" and omit the mention of concentration units.

40: And the magnitude of the inflow itself, so perhaps better to say "inflow conditions." Thank you for your comment. We now write simply "inflow."

34-36: Another common technique is to define background concentrations via statistical analysis of the lowest concentrations observed by the measurements used for the inversion.

Thank you for your suggestion. We've clarified our description of the second method to encompass more broadly methods that are directly data driven:

"BCs can be provided by coarse-resolution global simulations (Göckede et al., 2010; Wecht et al., 2014), by statistical or meteorological analysis of trace gas observations (Lauvaux et al., 2016; Balashov et al., 2020), or by combining simulated and observed concentrations (Sargent et al., 2018; Estrada et al., 2024)

91: (DOFS), i.e., the number of...

Thank you for your suggestion. We believe the phrasing is clear without adding "i.e.".

98: as the diagnostic that estimates

We made this change.

102: It's still not particularly clear what you mean by these two different methods, but maybe that will come shortly.

We hope that the description of these two methods in the remainder of this paragraph clarified the confusion. We added the following text to clarify the goal of these methods: "We also consider the

effect of inverse methods to decrease BC biases within the framework of the constant parameter, one-dimensional inverse model. We compare the effect of optimizing the BC concentrations (boundary method) or fluxes outside of the domain of interest (buffer method)."

123: the change in the influence —> the influence Thank you for the suggestion. We deleted this section.

173: What equations are used to calculate the error for the intermediate case? We now clarify that we calculate this numerically using Eq. (5).

196: p..

We corrected this.

198: Where does Eq 10 come from?

Equation (10) comes from the inversion of a single grid cell as described in the preceding paragraph. To clarify this, we now state: "We derive the conditions under which these methods are equivalent by considering an inversion of a single grid cell."

391: Is this using Eq 9, 13 or 15? We clarified that we are using the preview, which is described by Eq. (12).

S36: You mean Eq 9?

Thank you for catching this. We have corrected the text accordingly.